Bone marrow mesenchymal stem cell-derived exosomes protect podocytes from HBx-induced ferroptosis

Yang Xiaoqian
Yu Yani
Li Baoshuang
Chen Yueqi
Feng Moxuan
Hu Yongzheng
Jiang Wei jiangwei866@qdu.edu.cn
Department of Nephrology, The Affiliated Hospital of Qingdao University , Qingdao , China
Navaneethabalakrishnan Shobana
Electronic publication date: 2023 May 11
Publication date: 2023
Volume: 11
Electronic Location ID: e15314
Received 2022 Dec 13; Accepted 2023 Apr 6
Copyright: ©2023 Yang et al.
Copyright year: 2023
Copyright holder: Yang et al.
License: This is an open access article distributed under the terms of the Creative Commons Attribution License, which permits unrestricted use, distribution, reproduction and adaptation in any medium and for any purpose provided that it is properly attributed. For attribution, the original author(s), title, publication source (PeerJ) and either DOI or URL of the article must be cited.
License URL: https://creativecommons.org/licenses/by/4.0/

Keywords: Exosomes, Ferroptosis, HBx, Bone marrow mesenchymal stem cells, miR-223-3p

Funding: National Natural Science Foundation of China 81870494 Chinese Society of Nephropathy 20010080800 Qingdao Outstanding Health Professional Development Fund 2020-2022 This study was supported by the National Natural Science Foundation of China (NSFC 81870494), the Chinese Society of Nephropathy (20010080800), and the Qingdao Outstanding Health Professional Development Fund (2020-2022). The funders had no role in study design, data collection and analysis, decision to publish, or preparation of the manuscript.

==============================
Introduction

Hepatitis B virus-associated glomerulonephritis (HBV-GN) is a common secondary kidney disease in China, the pathogenesis of which is not completely clear, and there is still a lack of effective treatment.

Methods

The mechanism of exosomes derived from bone marrow mesenchymal stem cells (BMSCs) was investigated by using HBx-transfected human renal podocytes. Cell viability was detected by CCK8 assay. Iron and malondialdehyde (MDA) contents were detected by using commercial kits. Reactive oxygen species (ROS) levels were measured by flow cytometry analysis. The expression of ferroptosis related molecules was detected by quantitative real-time polymerase chain reaction (qRT-PCR) and Western blot. The effect of miR-223-3p transferred by BMSC-derived exosomes on HBx-overexpressing podocytes was proved by using miR-223-3p inhibitor.

Results

The cell viability of podocytes reduced at 72 h or 96 h after the transfection of lentivirus overexpressing HBx protein (p < 0.05). Ferroptosis-related proteins, including glutathione peroxidase 4 (GPX4) and solute carrier family 7 member 11 (SLC7A11) were down-regulated upon HBx overexpression, while acyl-CoA synthetase long-chain family member 4 (ACSL4) was up-regulated (p < 0.05). Intracellular levels of iron, MDA, and ROS were also enhanced (p < 0.05). BMSC-derived exosomes protected against ferroptosis induced by HBx overexpression in podocytes. miR-223-3p was enriched in BMSC-derived exosomes. Application of miR-223-3p inhibitor reversed the protective effect of BMSC-derived exosomes on HBx-induced ferroptosis in podocytes.

Conclusion

BMSC-derived exosomes inhibit HBx-induced podocyte ferroptosis by transferring miR-223-3p.

Introduction

Hepatitis B virus-associated glomerulonephritis (HBV-GN) remains a huge healthcare burden worldwide, although the prevalence of hepatitis B virus (HBV) infection has declined due to the vaccination policy (Hu et al., 2020; Liu et al., 2020). The prevalence of chronic HBV infection was 4.1% globally and 7.8% in China (Collaborators GBDHB, 2022). Among those patients in China, about 6.8–20% may progress into HBV-GN (Liu et al., 2020). It is a common secondary renal disease in most developing countries including China (Yang et al., 2019). The most frequent subtype is membranous nephropathy (MN) which usually presents with nephrotic syndrome clinically (Hu et al., 2020). Some patients may develop mild to moderate proteinuria accompanied by hematuria, and finally progress to end-stage of renal disease (ESRD). HBx as a multifunctional protein encoded by HBV can modulate multiple signaling pathways such as HBV replication, cellular transcription, cell cycle, DNA repair, etc (Lei et al., 2019; Yang et al., 2018). Past studies have shown that mutations at key genetic loci plays an important role in HBV-GN progression (Hui et al., 2014). HBx overexpression could decrease cell viability of podocytes and cause increased podocyte injury (Lei et al., 2019). Podocytes function as an essential component of glomerular filtration barrier, whose impairment leads to proteinuria, while a reduced number of podocytes is considered to be a relative risk factor for progressive renal impairment (He et al., 2017). Thus, we consider that podocytes injury induced by HBx overexpression may be an important pathological process in the development of HBV-GN, however the exact molecular mechanism remains unclear.

Ferroptosis is a form of programmed cell death characterized by iron dependence and accumulation of lipid peroxides (Li et al., 2020a; Xu et al., 2021). Excessive iron produces a large number of reactive oxygen species (ROS) via lipid peroxidase or non-enzymatic free radical reaction, which can synthesize lipid peroxides together with unsaturated fatty acids of lipid membranes, leading to membrane damage and cell death (Xu et al., 2021). Studies have found that increased lipid peroxides in podocytes cultured with high glucose could induce ferroptosis in podocytes (Wu et al., 2022b). Ferroptosis inhibitors could remarkably reduce iron content in high fructose-cultured podocytes, inhibit podocyte ferroptosis and alleviate diabetic kidney damage (Zhang et al., 2021). Therefore, we speculate that podocyte ferroptosis may play an essential role in renal disease pathogenesis, while restraining podocyte ferroptosis could slow the progression of kidney disease. Recent studies have shown that elevated level of iron in hepatocytes due to HBx overexpression could induce hepatocyte ferroptosis of acute liver failure by triggering lipid peroxidation (Liu et al., 2021a). However, the underlying mechanism by which ferroptosis acts during the development of HBV-GN requires clarification.

Mesenchymal stem cells (MSCs) are a kind of pluripotent stem cells that possess multiple differentiation potentials (Nikfarjam et al., 2020). It has been shown that MSCs could be used to treat kidney disease, and the key lies in releasing extracellular vesicles (EVs) via paracrine(Bochon et al., 2019; Wu et al., 2022a). Exosomes are one of the main subpopulations of EVs, containing a wide variety of proteins and miRNAs that take part in regulating numerous physiological and pathological processes (Lin et al., 2022). Previous studies have found that the injection of exosomes derived from bone marrow mesenchymal stem cells (BMSCs) into diabetic nephropathy rats can significantly improve the oxidative stress injury of renal tissue, reduce the discharge of urine protein and protect kidney function (Mao, Shen & Hu, 2021). This indicates that the exosomes are important for MSCs’ protective role in the treatment of kidney disease. In addition, recent studies found that MSC-derived exosomes were also involved in the regulation of ferroptosis (Lin et al., 2022). Increasing evidence have revealed that exosomal miRNA cargo is largely responsible for the protective effects of these exosomes (Ding et al., 2020; Yi & Tang, 2021). MiR-223-3p is one of the most abundant miRNAs in BMSCs (Zhao et al., 2020). Lu et al. (2019) found that MSC-derived exosomes could alleviate liver injury in acute liver failure rats by delivering miR-223-3p. It has also been reported that the deficiency of miR-223 may be closely related to the exacerbation of lupus nephritis (Hiramatsu-Asano et al., 2020). This suggests exosomal miR-223-3p is critical to tissue injury improvement. However, the practical effect of BMSC-derived exosomes and its miRNAs on podocyte injury in HBV-GN are still unclear. In this study, we hypothesize that BMSC-derived exosomes are key regulatory factors in the pathogenesis of HBV-GN by regulating podocyte ferroptosis, in which miR-223-3p plays a vital role. This study may provide a new direction for the cognition and treatment of HBV-GN.

Materials and Methods

Cell culture and transfection

Human renal podocytes were obtained from BeNa Culture Collection (Henan, China), and were cultured in Dulbecco’s Modified Eagle Medium (DMEM) containing 10% fetal calf serum at 37 °C with 5%CO2, and saturated humidity. Negative control lentivirus, HBx overexpression lentivirus, and miRNA inhibitor were synthesized by Biogenetech (Anhui, China). Podocytes were inoculated on 6-well plates at 5*104 cells/cm2. When reached 70% confluence, those podocytes were transfected with negative control lentivirus (lenti-NC), HBx overexpressed lentivirus (lenti-HBx), or nothing (control) according to the manufacturer’s protocol. The partial experimental groups were as follows: the control group, lenti-NC group, lenti-HBx group. Following transfection of podocytes for 72 h, the cells were observed by inverted fluorescence microscopy system (NSHOT) and measured the efficiency of lentivirus transfection into target cells.

In the exosome experiment, podocytes were treated with exosomes with protein concentration of 0.4 µg/µL for 48 h after lentivirus transfection. In the miR-223-3p inhibitor experiment, lipofectamine 2000 (Invitrogen, Waltham, MA, USA) was used to transfect podocytes. In brief, after treated with exosomes, cells were inoculated into 6-well plates for culture, when reached 50% degree of fusion, the cells were transfected with 5 µL of inhibitor NC or miR-223-3p inhibitor for 48 h. The transfection efficiency was examined using qRT-PCR. miR-223-3p inhibitor incorporates the reverse complementary synthetic target sites specific binding those target miRNAs and are specially modified, making miRNAs unavailable for normal function.

CCK-8 assay

Cell viability was detected by Cell Counting Kit-8 assay. Briefly, differently treated podocytes were inoculated into 96-well plates at 3,000 cells/well. Then, 10 µL CCK-8 reagent (Servicebio, Wuhan, Hubei, China) was added to each well and incubated for 2 h. Each well’s optical density (OD) at 450 nm was detected by a microplate reader (Thermo Fisher Scientific, Waltham, MA, USA) to reflect the cell viability.

Measurement of MDA and iron

Cells from each group were collected and then centrifuged to remove floating cells and cell fragments. After the cells were broken up by ultrasound treatment and subjected to centrifugation to obtain the supernatant, protein concentration was determined using BCA Assay Kit (Beyotime Biotechnology, Shanghai, China). Malondialdehyde (MDA) quantification was performed using MDA Assay Kit (Sinovac Biotech, Beijing, China) according to the manufacturer’s instructions. Briefly, 200 µL sample or distilled water was incubated with an MDA reagent for 60 min at 100 °C. After cooled and centrifuging, the absorbance was measured at 532 nm and 600 nm with a spectrophotometer for calculating MDA content. Similarly, Fe2+ quantification was performed using Iron Assay Kit (Solarbio, Beijing, China) according to the manufacturer’s instructions. 400 µL sample or standard or distilled water was incubated with Fe2+ reagent for 5 min at 100 °C. After mixing with 200 µL chloroform and centrifuged, the absorbance was measured at 532 nm and 600 nm with a spectrophotometer for calculating Fe2+ content.

Flow cytometry analysis

Reactive oxygen species (ROS) were detected by flow cytometry analysis. According to the manufacturer’s protocol of ROS Assay Kit (Meilunbio, Dalian, Liaoning, China; cat. no. MA0219), podocytes of each group were incubated with 10 µmol/L DCFH-DA probe or PBS for 60 min at 37 °C. The cell suspension was collected and centrifuged at 1,000 rpm/min for 5 min. After washing and centrifuging with PBS, cell precipitates were collected. Finally, the mean fluorescence intensity of each group was detected using a flow cytometer (Agilent, Santa Clara, CA, USA) to reflect the corresponding ROS level.

Western blot

Protein exacted from podocytes were prepared in RIPA lysis buffer (Boster Biological Technology, Wuhan, Hubei, China), and were estimated using BCA protein assay kit (Beyotime). Proteins were separated by SDS-PAGE (G2003, Servicebio), transferred onto PDVF membranes (Millipore) after electrophoresis, and blocked by 5% skimmed milk at room temperature for 1 h. After incubation with primary antibodies at 4 °C overnight, the membranes were incubated with secondary antibodies. We quantified the band area using AlphaEase FC software (Alpha Innotech) to reflect the relative expression of target proteins. The results were corrected to the band intensity of GAPDH. The antibodies used are shown in Table 1.

Table 1 Antibodies list.

Antibody	Vendor	Catalog no.	Working dilution	
GPX4	Proteintech	14432-1-AP	1:1000	
SLC7A11	Proteintech	26864-1-AP	1:1000	
ACSL4	ABclonal	A6826	1:1000	
CD9	Abcam	ab92726	1:1000	
TSG101	Abcam	ab125011	1:1000	
Calnexin	Abcam	ab22595	1:1000	
GAPDH	Proteintech	60004-1-Ig	1:5000	
Notes.

GPX4 glutathione peroxidase 4

SLC7A11 solute carrier family 7 member 11

ACSL4 acyl-CoA synthetase long-chain family member 4

qRT-PCR

Total RNAs were isolated from podocytes using TRIzol reagent (Cwbio, Jiangsu, China). Reverse transcription was conducted using Evo-M-MLV Reverse Transcriptase (Accurate Biology, Hunan, China). Subsequently, qPCR reaction was performed using SYBR Green Pro Taq HS premixed qPCR Kit (Accurate Biology) according to the manufacturer’s instructions, while the annealing temperature of the primers was set at 60 °C. GAPDH served as an internal control. The relative expression of target genes was calculated using the 2−ΔΔCt method and normalized by GAPDH expression. The primer sequences used are shown in Table 2. The primer sequence of miR-223-3p was provided by Ribobio (Guangzhou, China), and U6 served as its internal control.

Table 2 Primers list.

Gene	Primer sequence (5′–3′)	
GPX4	F, GCTGGACGAGGGGAGGAG	
	R, GGAAAACTCGTGCATGGAGC	
SLC7A11	F, TCTCCAAAGGAGGTTACCTGC	
	R, AGACTCCCCTCAGTAAAGTGAC	
ACSL4	F, ACTGGCCGACCTAAGGGAG	
	R, GCCAAAGGCAAGTAGCCAATA	
GAPDH	F, AGAAGGCTGGGGCTCATTTG	
	R, AGGGGCCATCCACAGTCTTC	
U6	F, GCTCGCTTCGGCAGCACA	
	R, GAACGCTTCACGAATTTGCGTG	
Notes.

F forward

R reverse

Isolation and characterization of BMSC-derived exosomes

Mouse bone marrow-derived MSCs were purchased from Procell (Wuhan, Hubei, China) and incubated in DMEM without fetal bovine serum (FBS), but with 10% fetal calf serum at 37 °C with 5% CO2 and saturated humidity. After reaching 70% confluence, the culture medium was centrifuged at 2,000 ×g at 4 °C for 30 min, and then filtered. The filtrate was centrifuged twice at 100,000 ×g at 4 °C for 70 min to remove floating cells and cell fragments, and stored at −80 °C. 10 µL exosome suspension drops were added for precipitation, and 3% phosphotungstic acid was added for stain. Scattering light analysis performed by NanoFCM particle size analyzer (Xiamen, Fujian, China) was used to measure the size of the collected exosomes. The morphology of exosomes was observed using the transmission electron microscope (Hitachi, Japan). Exosomes were dissolved in radioimmunoprecipitation assay buffer and quantified using BCA Assay Kit (Beyotime). CD9, TSG101, and Calnexin detected by Western blotting were also used to identify the collected exosomes. Using 293T cell lysate as positive control to eliminate experimental operation problems. The antibodies used are shown in Table 1.

Uptake test of BMSC-derived exosomes

To determine the uptake of BMSC-derived exosomes by human podocytes, exosomes were labeled with red fluorescent dye PKH26 (Umibio, Shanghai, China) and incubated with podocytes in an incubator for 2 h, 6 h and 24 h. Following PBS rinsing, cells were fixed with 4% paraformaldehyde for 30 min, washed with PBS, and then stained with 4′,6-diamidino-2-phenylindole. The differential interference contrast (DIC) microscopy image and fluorescence microscopy image were taken by Leica SP8 confocal microscope (Leica Microsystems). And the exosome uptake efficiency was expressed by the proportion of PKH26-positive podocytes at 2 h, 6 h and 24 h.

Statistical analysis

Data analysis was performed using GraphPad Prism 9.0 software (GraphPad Software, Inc., La Jolla, CA, USA). All data were expressed as mean ± standard deviation. Data among groups were compared using Student’s t-test or one-way ANOVA. p < 0.05 was deemed statistically significant.

Results

HBx overexpression induced podocytes ferroptosis

To study the mechanism of HBx-induced podocyte injury, we transfected human renal podocytes with HBx-overexpressed lentivirus and two control groups were set. Cell viability was detected using CCK-8 assay at 24 h, 48 h, 72 h, and 96 h, respectively. As shown in Fig. 1A, compared with control groups, the cell viability of human renal podocytes significantly decreased at 72 h (p =0.049) or 96 h (p =0.020) after the transfection of HBx-overexpressed lentivirus. Then, we detected the effect of HBx overexpression on ferroptosis via assessing the lipid peroxidation product MDA and iron levels after lentivirus transfection for 72 h. As presented in Figs. 1B and 1C, HBx remarkably increased the intracellular concentration of MDA (p =0.012) and iron (p =0.009) compared to that of the control cells. Flow cytometry was used to examine the reactive oxygen species generation (Fig. 1D). The results revealed that compared with either control or lenti-NC group, the level of ROS was significantly upregulated in lenti-HBx group (p < 0.001). Finally, the expression of ferroptosis-related molecules glutathione peroxidase 4 (GPX4), acyl-CoA synthetase long-chain family member 4 (ACSL4), and solute carrier family 7 member 11 (SLC7A11) were detected by Western blot and qRT-PCR (Figs. 1E, 1F). The results indicated that HBx reduced GPX4 (pwesternblot = 0.011; pqRT−PCR = 0.002) and SLC7A11 (pwesternblot = 0.023; pqRT−PCR<0.001), but induced ACSL4 (pwesternblot = 0.022; pqRT−PCR = 0.001) expression at both protein and mRNA levels. These findings suggested that HBx triggered ferroptosis in podocytes.

Figure 1 HBx overexpression induced podocytes ferroptosis.

(A) The cell viability of podocytes as detected by CCK-8 assay. (B) The concentration of iron in podocytes. (C) The concentration of MDA in podocytes. (D) The ROS generation as detected by flow cytometry. (E) The protein expression of ferroptosis related proteins (GPX4, ACSL4, SLC7A11) in podocytes as detected by Western blot analysis, normalized to GAPDH. (F) The mRNA expression of GPX4, ACSL4, SLC7A11 in podocytes as detected by qRT-PCR. The experiment was repeated three times. Control, podocytes without any transfection, Lenti-NC, podocytes transfected with negative control lentivirus, Lenti-HBx, podocytes transfected with HBx overexpression lentivirus. *, p < 0.05, **, p < 0.01, ***, p <0.001 versus Control or Lenti-NC.

Characterization of exosomes isolated from BMSCs

To better elucidate the effect of exosomes on podocyte injury, exosomes were isolated from BMSCs. Firstly, exosomes were observed as cup-shaped or spherical vesicles enveloped with lipid bilayer under transmission electron microscope (Fig. 2A). Next, scattering light analysis indicated that exosomes ranged in diameter from 30 to 150 nm, and the mean diameter was 78.41 nm (Fig. 2B). Finally, Western blot analysis of exosomal surface markers verified the expression of TSG101 and CD9, whereas calnexin was absent (Fig. 2C). These results indicated the successful isolation of exosomes from BMSCs.

Figure 2 Characterization of exosomes isolated from BMSCs.

(A) Observation of morphology of exosomes by transmission EM (scale bar, 200 nm). (B) Size distribution of BMSC-derived exosomes measured by NTA analysis. (C) CD9, TSG101, and calnexin detected by Western blot analysis. CL, positive control; Exo, exosomes.

BMSC-derived exosomes inhibited podocyte ferroptosis induced by HBx

The ability of human renal podocytes to internalize exosomes from BMSCs was assessed using red fluorescence (PKH26) labeled exosomes. The presence of red fluorescence in podocytes indicated that podocytes were able to internalize BMSC-derived exosomes (Fig. 3A). This provided the basis for the action of exosomes in podocytes. We also found that the exosome uptake efficiency by podocyte increased in a time-dependent manner, which reached 100% at 24 h. After HBx-induced cell ferroptosis, podocytes were co-incubated with BMSC-derived exosomes. The results showed that cotreatment with HBx and exosomes improved the cell viability of podocytes at 72 h (p =0.001) or 96 h (p < 0.001) in comparison with HBx treatment alone (Fig. 3B). Then the detection of ferroptosis-related markers (Figs. 3C and 3D) indicated that cotreatment of HBx and exosomes reduced the content of Fe2+ (p =0.010) and MDA (p =0.070) in podocytes versus the lenti-HBx group. Flow cytometry (Fig. 3E) suggested that exosome treatment could significantly lower the ROS level in podocytes transfected with HBx (p < 0.001). Western blot analysis was used to detect the expression of ferroptosis-related proteins in podocytes (Fig. 3F). The results suggested the cotreatment of HBx and exosomes could increase the expression of GPX4 (p =0.018) and SLC7A11(p < 0.001), but decrease the expression of ACSL4 (p =0.030) at the protein level. Also, qRT-PCR detection (Fig. 3G) found the expression of these molecules at mRNA level appearing the same trend (pGPX4 = 0.004, pSLC7A11 <  0.001, pACSL4<0.001). Taken together, these data demonstrated that exosomes from BMSCs were capable of suppressing HBx-induced ferroptosis of podocytes.

Figure 3 BMSC-derived exosomes inhibited podocyte ferroptosis induced by HBx.

(A) Differential interference contrast (DIC) microscopy image and fluorescence microscopy image taken by laser scanning confocal microscope (scale bar 25 µm, 24 h). Red fluorescence PKH26 used to label BMSC-derived exosomes. Uptake efficiency of PKH26-labeled exosomes by podocytes. (B) The cell viability of podocytes as detected by CCK-8 assay. (C) The concentration of iron in podocytes. (D) The concentration of MDA in podocytes. (E) The ROS generation as detected by flow cytometry. (F) The protein expression of ferroptosis related proteins (GPX4, ACSL4, SLC7A11) in podocytes as detected by Western blot analysis, normalized to GAPDH. (G) The mRNA expression of GPX4, ACSL4, SLC7A11 in podocytes as detected by qRT-PCR. The experiment was repeated three times. Control, podocytes without any transfection; Lenti-NC, podocytes transfected with negative control lentivirus; Lenti-HBx, podocytes transfected with HBx overexpression lentivirus; Lenti-HBx+exo, podocytes treated with exosomes after transfected with HBx overexpression lentivirus. *, p < 0.05, **, p < 0.01, ***, p < 0.001 versus Control or Lenti-NC. #, p < 0.05, ##, p < 0.01, ###, p < 0.001 versus Lenti-HBx.

Exosomes from BMSCs delivered miR-223-3p to suppress HBx-induced ferroptosis in podocytes

The expression of miR-223-3p in HBx-induced podocytes was measured by qRT-PCR after treatment with BMSC-derived exosomes. miR-223-3p expression was obviously lower after induction with HBx than the control group (p < 0.001), but it was significantly increased by the addition of BMSC-derived exosomes containing miR-223-3p (p =0.009). To further determine the effect of miR-223-3p transferred by BMSC-derived exosomes on HBx-overexpressing podocytes, HBx-overexpressing podocytes were transfected with miR-223-3p inhibitor. It seemed that treatment with miR-223-3p inhibitor caused a decline (p =0.001) in miR-223-3p expression (Fig. 4A). Next, the CCK8 assay (Fig. 4B) illustrated that the treatment with BMSC-derived exosomes and miR-223-3p inhibitor remarkably attenuated cell viability of podocytes at 72 h (p < 0.001) or 96 h (p < 0.001). The treatment with miR-223-3p inhibitor reversed the downregulation of iron (p < 0.001), MDA (p < 0.001), and ROS (p < 0.001) in HBx-overexpressing podocytes mediated by BMSC-derived exosomes (Figs. 4C–4F). In addition, qRT-PCR and Western blot analysis (Figs. 4G and 4H) revealed that miR-223-3p inhibitor weakened the effects of BMSC-derived exosomes on GPX4 (pwesternblot < 0.001, pqRT−PCR = 0.003) and ACSL4 (pwesternblot < 0.001, pqRT−PCR < 0.001) expression at both protein and mRNA levels. However, though the treatment with miR-223-3p inhibitor attenuated the upregulation of SLC7A11 expression caused by BMSC-derived exosomes at the protein level (p < 0.001), no significant statistical difference was observed at the mRNA level (p = 0.107). Collectively, these results indicated that delivery of miR-223-3p into podocytes by BMSC-derived exosomes suppressed HBx-induced ferroptosis.

Figure 4 Exosomes from BMSCs delivered miR-223-3p to suppress HBx-induced ferroptosis in podocytes.

(A) The expression of miR-223-3p as detected by qRT-PCR. (B) The cell viability of podocytes as detected by CCK-8 assay. (C) The concentration of iron in podocytes. (D) The concentration of MDA in podocytes. (E–F) The ROS generation as detected by flow cytometry. (G) The protein expression of ferroptosis related proteins (GPX4, ACSL4, SLC7A11) in podocytes as detected by Western blot analysis, normalized to GAPDH. (H) The mRNA expression of GPX4, ACSL4, SLC7A11 in podocytes as detected by qRT-PCR. The experiment was repeated three times. Control, podocytes without any transfection; HBx, podocytes transfected with HBx overexpression lentivirus; HBx+exo, podocytes treated with exosomes after transfected with HBx overexpression lentivirus; HBx+exo+inhibitor NC, podocytes treated with exosomes and negative control miRNA inhibitor after transfected with HBx overexpression lentivirus, HBx+exo+miR-223-3p inhibitor, podocytes treated with exosomes and miR-223-3p inhibitor after transfected with HBx overexpression lentivirus. ***, p < 0.001 versus Control. #, p < 0.05, ##, p < 0.01, ###, p < 0.001 versus HBx. &, p < 0.05, &&, p < 0.01, &&&, p < 0.001 versus HBx+exo+inhibitor NC.

Discussion

HBV-GN has been identified as the most common extrahepatic lesion caused by HBV infection (Yang et al., 2019). Podocyte injury may be an important factor in the pathogenesis of HBV-GN (He et al., 2017; Lei et al., 2019). In this study, we found that ferroptosis played a critical role in HBx-induced podocyte injury. In addition, we also found that BMSC-derived exosomes inhibited HBx-induced podocyte ferroptosis through miR-223-3p.

Ferroptosis is identified as a form of regulated cell death that depends on iron metabolism and lipid peroxidation (Xu et al., 2021). Studies have suggested that ferroptosis was related to the pathogenesis of multiple renal diseases, including acute kidney injury (AKI) and chronic kidney diseases such as diabetic nephropathy (Guan et al., 2021; Mishima et al., 2020). Iron-dependent ROS accumulation is considered to be a trigger of ferroptosis: excessive iron activates the lipid peroxidation chain reaction by participating in the generation of ROS to produce massive lipid peroxides, which result in membrane damage and cell death (Kajarabille & Latunde-Dada, 2019). Thus, iron and lipid ROS levels can be used to evaluate the presence of ferroptosis. Malondialdehyde (MDA), as a common by-product of lipid peroxidation, allows the measurement of lipid peroxidation (Xu et al., 2021). The level of MDA was found to raise in folic acid-induced AKI mice kidney tissue (Wang et al., 2021). Similarly, the level of ROS improved in renal tubular epithelial cells of mice with AKI caused by ischemia-reperfusion injury, which indicated that ferroptosis contributed significantly to the process of renal tubular necrosis in AKI (Wang et al., 2021). Zhang et al. (2021) found that high glucose-induced ferroptosis of podocytes, which was manifested as increased levels of iron, ROS, and MDA. The study also proved that ferroptosis activator erastin reduced podocyte viability. HBx can elevate intracellular ROS levels and participate in mitochondrial oxidative stress (Li et al., 2022). Recent studies supported the critical role of mitochondrial oxidative stress in ferroptosis (Gan, 2021). In this study, we found that the viability of podocytes transfected with HBx at 72 h or 96 h was markedly lower than the two control groups, though there was no significant difference in cell viability between experimental groups at 24 h or 48 h. Furthermore, the overexpression of HBx obviously enhanced the levels of iron, MDA, and ROS in podocytes. Researchers have found that ACSL4 was also involved in the lipid peroxidation of polyunsaturated fatty acids by generating acyl-CoA (Cui et al., 2021). Normally GPX4 is capable of metabolizing lipid peroxides to non-toxic lipid alcohols at the expense of glutathione (GSH) (Xu et al., 2021). The biosynthesis of important intracellular antioxidant GSH requires the participation of cysteine, which is produced from cystine transported by the cell surface cystine/glutamate antiporter system Xc− (Li et al., 2020a). SLC7A11 is the active subunit of system Xc−. Regulating the expression of SLC7A11 will affect the uptake of cystine, leading to the depletion of GSH and ferroptosis (Wu et al., 2022b). Researchers found that the level of GSH as well as the expression of GPX4 and SLC7A11 have decreased during podocyte ferroptosis induced by high fructose (Wu et al., 2022b). Similarly, another research detected that high glucose-induced podocyte ferroptosis, decreased cellular GPX4, and increased ROS content and ACSL4 expression in diabetic nephropathy (Jin et al., 2022). In acute liver failure, HBx induced hepatocyte ferroptosis by activating the SLC7A11 inhibition pathway to weaken the expression of SLC7A11, GSH, and GPX4, and to potentiate intracellular iron deposition (Liu et al., 2021a). Interestingly, Kuo et al. (2020) found that HBx could inhibit hepatic stellate cell ferroptosis and lead to liver fibrosis. In this study, we found that overexpression of HBx in podocytes downregulated the expression of antioxidant GPX4 and SLC7A11, but upregulated the expression of ACSL4 which was related to lipid oxidation. These findings verify the crucial role of HBx-induced podocyte ferroptosis in HBV-GN.

At present, antivirus therapy has been the backbone of HBV-GN treatment due to its efficacy in HBV clearance and proteinuria remission, but fails to improve the glomerular filtration rate (Fu et al., 2020). Therefore, there is an urgent need for more effective treatments to reduce the occurrence and progression of HBV-GN. MSCs can be isolated from adult tissues including bone marrow, adipose tissue, umbilical cord blood, and amniotic fluid, which are easy to obtain and considered an ideal choice for cell therapy (Nikfarjam et al., 2020). Research have proved that most of the MSC-related beneficial effects can be attributed to the effectiveness of released exosomes. Whereas, compared with cell treatment, exosomes have better plasticity which means we can obtain effective target products through exogenous modification (Wu et al., 2022a). Moreover, embolism, tumor proliferation, immune response, and other risks can be avoided (Lu et al., 2019; Wu et al., 2022a). Currently, MSC-derived exosomes have been demonstrated to play similar therapeutic effects as ferroptosis inhibitor Fer-1 in regulating lipid ROS accumulation and ferroptosis-related gene expression (Lin et al., 2022). For instance, MSC-derived exosomes significantly attenuated CCl4-induced lipid peroxidation in hepatocytes, which is manifested by reduced MDA and lipid ROS in cells, and added SLC7A11 expression (Lin et al., 2022). Exosomes produced by MSCs can reduce the content of MDA in ischemia-reperfusion kidney tissue, increase the level of antioxidant enzyme SOD, and reverse the decline of podocyte viability induced by high glucose(Jin et al., 2019; Li et al., 2020b). Our study manifested that BMSC-derived exosomes significantly attenuated HBx-provoked lipid peroxidation, as manifested by the levels of MDA and ROS in podocytes, and improved podocyte viability. Moreover, BMSC-derived exosomes also abolished other ferroptosis biomarkers raised by HBx overexpression, including decreased expression of GPX4 and SLC7A11 and increased expression of ACSL4. Zhang et al. (2016) found that extracellular vesicles derived from MSCs activated the Nrf2/ARE pathway to upregulate the level of antioxidant enzymes, and to reduce the level of oxidative stress damage factor MDA, thus achieving renal injury improvement. This was consistent with our findings. Thus, it could be concluded that BMSC-derived exosomes effectively alleviate HBx-induced podocyte ferroptosis.

As we all known, the functional significance of exosomes is largely lie in its content: miRNAs, mRNAs and proteins. As the most important exosomal component, miRNAs can be selectively encased into exosomes and be involved in the regulation of ferroptosis. Ding et al. (2020) found that ischemia-reperfusion injury induced the upregulation of miR-182-5p and miR-378a-3p, and activated renal ferroptosis by downregulating GPX4 and SLC7A11. Human umbilical cord MSC-derived exosomes inhibited the deposition of iron and MDA in cardiomyocytes by carrying miR-23a-3p to protect cardiomyocytes from ferroptosis induced by ischemia-reperfusion (Song et al., 2021). Previous studies have found that miR-223-3p was highly encapsulated in exosomes released from BMSCs, which was also proved by our study (Zhao et al., 2020). miR-223-3p can reduce brain ischemia-reperfusion injury by inhibiting inflammatory response, and also promote uterine endothelial angiogenesis to repair acute uterine injury (Liu et al., 2021b; Zhao et al., 2020). In our study, we revealed that miR-223-3p delivered from BMSC-derived exosomes to podocytes suppressed podocyte ferroptosis and restored podocyte viability. Previous research has shown that overexpression of miR-223 decreased HDAC2 expression (Leuenberger et al., 2016). Silencing HDAC2 has been proved to reduce STAT3 phosphorylation, while STAT3 phosphorylation could affect the pathological process associated with cell ferroptosis (Pang et al., 2011; Qiang et al., 2020). Therefore, we hypothesized that miR-223-3p may affect podocyte ferroptosis in HBV-GN through the regulation of STAT3 phosphorylation by HDAC2; the specific mechanism needs further study. However, one limitation of our study was the lack of animal experimental verification. In addition, there was no significant statistical difference in SLC7A11 expression at the mRNA level after the treatment of the miR-223-3p inhibitor, though corresponding protein expression was memorably reduced (p < 0.001). The mismatching of transcript expression data and protein expression data may be caused by the following aspects: first, gene transcription and translation exist in time and site spacing; second, there will be corresponding processing or modification after transcription and translation; third, mRNA and protein own different half-lives, so the detection time may also affect the final data; fourth, there may be a certain amount of proteins reserve in cells that do not function.

Conclusion

In conclusion, our findings demonstrate that HBx overexpression induces podocyte ferroptosis and plays a crucial role in the course of HBV-GN. BMSC-derived exosomes protect podocytes from HBx-induced ferroptosis, and the main point is the delivery of miR-223-3p by exosomes. This provides a new research direction for the cognition and treatment of HBV-GN.

Additional Information and Declarations

Competing Interests

Author Contributions

Data Availability

Supplemental Information

Supplemental Information 1 Raw data

Click here for additional data file.

Supplemental Information 2 Western blot pictures

Click here for additional data file.

The authors declare there are no competing interests.

Xiaoqian Yang performed the experiments, prepared figures and/or tables, authored or reviewed drafts of the article, and approved the final draft.

Yani Yu performed the experiments, authored or reviewed drafts of the article, and approved the final draft.

Baoshuang Li analyzed the data, prepared figures and/or tables, and approved the final draft.

Yueqi Chen analyzed the data, prepared figures and/or tables, and approved the final draft.

Moxuan Feng analyzed the data, prepared figures and/or tables, and approved the final draft.

Yongzheng Hu analyzed the data, prepared figures and/or tables, and approved the final draft.

Wei Jiang conceived and designed the experiments, authored or reviewed drafts of the article, and approved the final draft.

The following information was supplied regarding data availability:

The raw data are available in the Supplemental Files.

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
