# Peer review of "Bone marrow mesenchymal stem cell-derived exosomes protect podocytes from HBx-induced ferroptosis"

_PeerJ, doi:10.7717/peerj.15314_

## Round 0.1 · original submission · Major Revisions

The manuscript has been assessed by three independent reviewers and I strongly suggest addressing the concerns raised by all three reviewers before your paper can be considered for publication. The manuscript is well written, and I appreciate the authors for including limitations in the discussion section. The authors are requested to elaborate the methods section. Figures in the results section need to be improved in terms of clarity and labeling. Inclusion of the chemical nature of the inhibitor and the mechanism of inhibition and the specificity is highly recommended.

Reviewer 1 ·

Basic reporting

No comments

Experimental design

No comments

Validity of the findings

No comments

Additional comments

The manuscript “Bone marrow mesenchymal stem cell-derived exosomes protect podocytes from HBx-induced ferroptosis” by Xiaoqian Yang et al, explores the potential role of BMSC derived exosomes in inhibiting HBx induced podocyte ferroptosis. The authors have shown than expression of HBx induces ferroptosis in podocytes and the effects can be alleviated by BMSC derived exosomes via delivery of miR-223-3p cargo. Overall, the identification of such potential targets can make significant contribution to drug discovery.
I do have some minor comments that authors can work upon and give a better explanation so that it is clear to the audience.

1: General comment: the authors should mention in the legend of the figures what does control, Lenti-NC, lenti-HBx, Hbx, Hbx+exo, and so on signify. Without proper explanation it becomes hard to interpret the results. Also, in some of the figures the p value which is denoted by (*) has been labelled as (#) or (&). Please correct this formatting error.
2: Fig 1B and 1C; also line 189-190: is the data show here from a 72h or 96h timepoint? The authors should mention that in the text.
3: Fig 1D: the gating strategy looks different for control, Lenti-NC and Lenti-HBx. For the mean intensity, there should be both positive and negative control.
4: line 128-135: kindly mention the cat no of the kit used for the assay to detect ROS.
5: Fig 3A: The authors should include the DIC or phase and a merged image of all of them for a better understanding of the localization of the PKH26 labelled BMSC derived exosome.
6: Fig 3E: The gating looks different for lenti-HBx. Can the authors explain why they chose to have a different gating in lenti-HBx?
7: what is the name of the inhibitor? If there are any refences, kindly add it to the text.
8: Fig 4C, 4D, 4F: kindly add the control data.

Reviewer 2 ·

Basic reporting

NA

Experimental design

NA

Validity of the findings

NA

Additional comments

See attached PDF

Annotated reviews are not available for download in order to protect the identity of reviewers who chose to remain anonymous.

Reviewer 3 ·

Basic reporting

In the present manuscript using in vitro studies, authors have demonstrated induction of podocyte ferroptosis upon HBx overexpression. Further they have shown that the treatment with BMSC-derived exosomes protects cells from HBx induced ferroptosis through the action of miR-223-3p.
Overall, the manuscript is well written, the language is easy to comprehend, but it needs to be polished further. There are some grammatical errors and missing words at some places, so authors are requested to read the manuscript carefully to correct those mistakes. The introduction section is well structured. The results have been discussed appropriately along with the limitations of present study.

Experimental design

-

Validity of the findings

-

Additional comments

Below are some concerns that need to be addressed:
• Results, Fig. 1D and 3E: In the scatter plot, why the gate applied to lenti-HBx group is not similar to the respective control group. You should show scatter plot or histogram overlays of different treatment groups for DCFH-DA signal.
• Results line no. 201: you can change the title as, ‘Characterization of exosomes isolated from BMSCs’.
• Fig.3A: It would be nice if authors can provide better images for exosome uptake with better cellular morphology and with more number of cells uptaking exosomes in the given representative image.
• In Fig.3F bar graph, there is an increase of ACSL4 and decrease of SLC7A11 in the lenti-NC treated group. What is the statistical significance of difference in this group with respect to control group? How do you explain the change in the levels of ACSL4 and SLC7A11 in this group?
• What are the treatment conditions for miR-223-3p inhibitor group as in Fig. 4E, HBx + exo+ miR-223-3p treatment significantly changes the side and forward scatter of the cells? Also show either the scatter plots or histograms of DCFH-DA stained cells for each group.
• What was the reason for selecting BM derived MSCs as source of exosomes compared to other sources of MSCs.
• Based on the existing literature, it would be nice if you could explain the mechanism of action of miR-223-3p in the discussion section.
• In the introduction section, mention the incidence rate of HBV and HBV-GN globally and in China.
• Methodology on co-treatment of podocytes with lenti-HBx and exosomes is missing. Similarly, the miR-223-3p inhibitor treatment conditions are also not described.
• In methods section, line no. 102: what do you mean by ‘partial’ experimental groups?
• Mention the annealing temperatures of qRT-PCR primers.

---

## Round 0.2 · Minor Revisions

The authors have addressed all the comments from the reviewers except the one pertaining to Fig 3A. As pointed by one of the reviewers, Fig3A is of high significance in the manuscript, it is essential to show the localization of the exosomes that have been uptake by the podocytes. The authors are asked to provide either phase image or IF images of any podocyte specific markers to show the cell morphology. The percentage of internalized exosomes needs to be provided in graphical or tabular form within the manuscript.

Reviewer 1 ·

Basic reporting

The authors have addressed all my questions except the fluorescence image (fig3A). Since that fig is of great significance in this paper, I would have loved to see the phase image being merged with the both DAPI and PKH26. Any fluorescent microscope should have the ability to take phase image if not DIC. Without that it is not possible to understand their localisation. I will strongly suggest the authors to retake this image along with the control or they should downgrade their findings and remove this image from the figure.

Experimental design

na

Validity of the findings

na

Additional comments

na

Reviewer 2 ·

Basic reporting

NA

Experimental design

NA

Validity of the findings

NA

Additional comments

NA

Reviewer 3 ·

Basic reporting

Authors have addressed all raised concerns satisfactorily. Therefore manuscript can be accepted for publication.

Experimental design

__

Validity of the findings

__

Additional comments

__

---

## Round 0.3 · accepted · Accept

The authors have adequately addressed all the comments and the manuscript is ready for acceptance in its present form.